# Molecular Analysis of the Complete Genome of a Simian Foamy Virus Infecting *Hylobates pileatus* (pileated gibbon) Reveals Ancient Co-Evolution with Lesser Apes

**DOI:** 10.3390/v11070605

**Published:** 2019-07-03

**Authors:** Anupama Shankar, Samuel D. Sibley, Tony L. Goldberg, William M. Switzer

**Affiliations:** 1Laboratory Branch, Division of HIV/AIDS Prevention, Center for Disease Control and Prevention, Atlanta, GA 30329, USA; 2Department of Pathobiological Sciences, School of Veterinary Medicine, University of Wisconsin-Madison, Madison, WI 53706, USA

**Keywords:** simian foamy virus, gibbon, lesser apes, co-evolution, complete viral genome

## Abstract

Foamy viruses (FVs) are complex retroviruses present in many mammals, including nonhuman primates, where they are called simian foamy viruses (SFVs). SFVs can zoonotically infect humans, but very few complete SFV genomes are available, hampering the design of diagnostic assays. Gibbons are lesser apes widespread across Southeast Asia that can be infected with SFV, but only two partial SFV sequences are currently available. We used a metagenomics approach with next-generation sequencing of nucleic acid extracted from the cell culture of a blood specimen from a lesser ape, the pileated gibbon (*Hylobates pileatus*), to obtain the complete SFVhpi_SAM106 genome. We used Bayesian analysis to co-infer phylogenetic relationships and divergence dates. SFVhpi_SAM106 is ancestral to other ape SFVs with a divergence date of ~20.6 million years ago, reflecting ancient co-evolution of the host and SFVhpi_SAM106. Analysis of the complete SFVhpi_SAM106 genome shows that it has the same genetic architecture as other SFVs but has the longest recorded genome (13,885-nt) due to a longer long terminal repeat region (2,071 bp). The complete sequence of the SFVhpi_SAM106 genome fills an important knowledge gap in SFV genetics and will facilitate future studies of FV infection, transmission, and evolutionary history.

## 1. Introduction

Foamy viruses (FVs) belong to the Retroviridae subfamily of Spumaretrovirinae, which have fundamentally different replication strategies compared to other complex retroviruses, such as human immunodeficiency virus (HIV). For example, FVs differ from other retroviruses in how they initiate infection. Like complex DNA viruses, FVs use two promoters for gene expression, one in the long terminal repeat (LTR) and one in the envelope (*env*) gene [1]. In addition, FVs complete reverse transcription within the virion before infection of a new host cell, such that the SFV genome can be double-stranded DNA or single-stranded RNA [2,3]. FVs infect a wide range of mammals, including cows, horses, cats, and nonhuman primates (NHPs) in which they are called simian foamy viruses (SFVs). SFVs have been identified in nearly every primate species examined across all continents where NHPs exist. Phylogenetic analyses show each of these NHPs to be infected with species-specific variants, reflecting a generally ancient co-evolution of SFVs and their hosts [3,4,5,6,7,8,9,10,11,12,13,14,15]. Like other simian retroviruses, SFVs can recombine and cross-species infections occur, both of which can complicate their evolutionary history [5,16,17,18,19]. Hence, analysis of complete genomes is necessary to fully understand the evolutionary trajectory of SFV.

SFVs received heightened public health attention following numerous reports of transmission of SFV from NHPs to humans across the globe via a variety of routes of exposure [3,12,20,21,22]. Many studies have documented SFV acquisition, both in persons working with NHPs in research facilities and zoos [20,21,23,24,25,26,27,28] and in humans exposed to NHPs in natural habitats, where hunting and butchering of primates and keeping NHPs as pets is common, especially in parts of Africa and Asia [24,25,26,27,29]. Although SFVs establish permanent infection of their primate hosts and in zoonotically infected humans, there has been no clear evidence of pathogenesis despite their ability to cause cytopathology in vitro [1,3,12,22,30,31,32,33,34]. Limited studies have also been unable to identify cases of person-to-person transmission [3,7,12,22,31]. As human populations expand and encroach upon NHP habitats, interaction among these species grows, increasing risks for SFV exposure and infection. Many areas of Asia, Africa, and South America have seen increases in deforestation with concomitant intensified incursions into NHP habitats [35,36]. The ensuing exposures to NHPs and their pathogens require continued monitoring, facilitated by the development and modification of new and existing assays for the detection of zoonotic viral agents, including SFV and other retroviruses.

The design of molecular methods for the accurate identification of SFV infection requires a database containing representative sequences from divergent SFV lineages. Although numerous partial nucleotide (nt) sequences (average length of about 500 nt, mostly in the polymerase (*pol*) gene), are available from a large number of SFVs from prosimians, Old World monkeys (OWMs), apes, and New World monkeys (NWMs), there remains a paucity of complete SFV genomes. Recently, the taxonomic classification of FVs has been standardized such that the SFV names include a lower case three-letter abbreviation, where the first letter is the first letter of the scientific name of the host genus and the next two letters are the first two letters of the species or subspecies they were isolated from [14]. Whole genome SFV sequences are currently available from one prosimian (SFVocr from a galago), four from NWMs (one each from a spider monkey (SFVaxx), marmoset (SFVcja), squirrel monkey (SFVssc), and capuchin (SFVsxa)), seven from OWMs (three from African green monkeys (SFVcae), four from macaques (SFVmcy, SFVmmu, SFVmfu, SFVmfa), one from a human infected with SFV from a spot-nosed guenon (SFVcni), and eight from great apes (four from chimpanzee (SFVpsc, SFVpve, SFVptr) species, including three from infected humans, three from gorilla (SFVggo), including two from infected humans, and one from an orangutan (SFVppy)) [14]. In contrast, there is a dearth of knowledge about SFV genomes of the smaller or “lesser” apes, family Hylobatidae (common name gibbons), despite their wide taxonomic diversity. Gibbons belong to the superfamily Hominoidea along with great apes and humans. Gibbons are found predominantly in tropical and sub-tropical forests of Southern and Southeast Asia from eastern Bangladesh and northeast India to southern China and Indonesia, including the islands of Sumatra, Borneo, and Java [37]. Unlike the great apes, which include about six species, gibbons are more diverse, with about 19 species identified depending on the classification source [38,39], most of which are endangered (www.iucnredlist.org) [40]. The Hylobatidae consists of five genera and 19 species (*Hoolock* species (*n* = 2), *Hylobates* sp. (*n* = 9), *Justitia* (*n* = 1), *Nomascus* sp. (*n* = 6), and *Symphalangus* sp. (*n* = 1).

A recent study reported a high seroprevalence of SFV in gibbons from Cambodia, although the seropositive samples were not SFV PCR-positive [41]. Like other NHPs, gibbons are frequently hunted or kept as pets, facilitating opportunities for human exposure to SFV; yet there is a lack of available sequences to optimize PCR assays for their detection. Considering these factors, elucidation of additional sequences from SFV-infected gibbons will provide the field with important molecular information for the development of diagnostic assays and will permit further examination of the biology and evolutionary history of SFV in apes, and in NHPs in general.

In a previous report [42], we described the isolation of a novel, highly divergent gibbon SFV strain (SFVhpi_SAM106) from a captive-born *Hylobates pileatus* (pileated gibbon) using blood cell co-culture, and the subsequent amplification of partial *pol* sequences. In this study, we used random hexamer-based deep-sequencing [43], a technique that uses random priming instead of relying on sequence-specific approaches, thus allowing molecular characterization of divergent viral sequences. In addition to *in silico* characterization of the full-length SFVhpi_SAM106 genome, we also analyzed evolutionary relationships to other complete monkey and ape SFVs by using non-simian FVs as outgroups.

## 2. Materials and Methods

### 2.1. Blood Sample Processing, Co-Culture, and PCR Identification of a Novel Divergent SFV in Gibbons

SFVhpi_SAM106 was previously isolated from peripheral blood mononuclear cells (PBMCs) prepared from whole blood from a *H. pileatus* (SAM106) co-cultured with canine thymocyte Cf2Th cells as described in detail elsewhere [42]. Briefly, frozen PBMCs were thawed, stimulated with interleukin-2 for three days at 37 °C, washed with media, and incubated with an equal number of Cf2Th cells. Cultures were monitored every 3 to 4 days for syncytial cytopathic effect (CPE) typical of FV. CPE was visible on day 38, whereupon infected Cf2Th cells and viral supernatants were collected and stored frozen in liquid nitrogen until use. This captive gibbon was wild-caught and was about 30 years old at the time of specimen collection. Previous PCR and sequence analysis of integrase sequences within the *pol* gene from this gibbon confirmed the presence of a highly divergent SFV distinct from other ape SFVs [42]. During that same study, we similarly isolated SFVhle from a *H. leucogenys* but were unable to recover virus from the stored tissue culture supernatants. As described in the initial publication, NHP blood specimens were obtained opportunistically in accordance with the animal care use committees at each participating institution [42].

### 2.2. Next Generation Sequencing and SFVhpi_SAM106 Genome Assembly

In total, 1 mL of tissue culture supernatant was centrifuged at 5000× *g* at 4 °C for 5 min with subsequent filtration through a 0.45 µm filter (Millipore, Billerica, MA, USA) to remove any residual host cells. Viral nucleic acids were then isolated using the Qiagen QIAamp MinElute virus spin kit (Qiagen, Hilden, Germany) according to the manufacturer’s instructions, except that carrier RNA was omitted. The eluted nucleic acids were then treated with DNase I (DNA-free, Ambion, Austin, TX, USA) and cDNA was generated by priming with random hexamers using the Superscript double-stranded cDNA Synthesis kit (Invitrogen, Carlsbad, CA, USA). cDNA was purified using the Agencourt Ampure XP system (Beckman Coulter, Brea, CA, USA) and approximately 1 ng of cDNA was subjected to simultaneous fragmentation and adaptor ligation (“tagmentation”) using the Nextera XT DNA Sample Prep Kit (Illumina Systems, San Diego, CA, USA). Briefly, fragmented DNA was PCR-amplified with Nextera index primers (15 cycles) and purified using the Agencourt Ampure XP system. The resulting DNA library was sequenced using an Illumina MiSeq (MiSeq Reagent Kit V3, 600 cycle, Illumina Systems, San Diego, CA, USA). To analyze sequence data, raw sequences were de-multiplexed and converted to FastQ format using the CASAVA v1.8.2 software (Illumina). The processed reads were then imported into the CLC Genomics Workbench v7 (CLC bio, Aarhus, Denmark), trimmed to remove Nextera-specific transposon sequences as well as short and low quality reads, and assembled using the CLC de novo assembler. Both singleton and assembled contiguous sequences (contigs) were queried against the GenBank database (http://www.ncbi.nlm.nih.gov/GenBank) using the basic local alignment search tools blastn and blastx [12], with a high e-value cut-off of 10 and word sizes of 11 and three for blastn and blastx queries, respectively. Contigs with significant homology to SFV were then mapped to SFVmcy (macaque SFV; GenBank accession number NC_010819) as the reference sequence to generate the consensus SFVhpi_SAM106 sequence.

### 2.3. Sanger Sequencing of LTR Region

We amplified the LTR region using primers specific to SFVhpi_SAM106 in order to confirm the LTR length. DNA was extracted from the tissue culture cells and 500 ng was used as template in two separate nested PCRs using SFVhpi_SAM106-specific primers. One assay spanned the 5′LTR and RU5 region while the second spanned the 3′end of *bet* and the 3′LTR. The primers for the SFVhpi_SAM106_LTR-RU5 and SFVhpi_SAM106_*bet*-LTR fragments for the primary and secondary PCRs respectively are:

SFVhpi_SAM106_LTRF2 (5′ GCAGTAGGAGAACAACCTCCTT 3′) and FVRU5R1 (5′ CCCGACTTATATTCGAGCCCCAC 3′) and

SFVhpi_SAM106_LTR_nestF2 (5′ GGAGGAATACTCCTCTCCCCCTCTC 3′); FVRU5R2 (5′ CACGTTGGGCGCCAAATTGTC 3′) and

SFVhpi_SAM106_*bet*_F1 (5′ GTGGGAGAAGGTAATATTAATCC 3′) and SFVhpi_SAM106_LTR_R1 (5′ GTGGAATATTCTGTGTTGATTATCC 3′) and

SFVhpi_SAM106_*bet*_nestF1 (5′ AGGCATATGGACCACCACAAG 3′) and SFVhpi_SAM106_LTR_nestR1 (5′ CAACCTTGTTGATAAGGGCAAC 3′).

We performed an initial denaturation at 94 °C for 2 min. followed by 40 cycles of 94 °C for 1 min, 45 °C for 1 min; 72 °C for 2.5 min, with a final extension at 72 °C for 5 min. For both nested PCRs, we used 3 µL of the primary PCR product as template. Nested PCR products were electrophoresed in 1.0% agarose gels and visualized by ethidium bromide staining. For sequence analysis, the PCR products were purified using the Qiaquick gel extraction kit (Qiagen Inc., Valencia, CA) and then sequenced in both directions using a Big Dye terminator cycle kit (ThermoFisher Scientific, Waltham, MA) and additional internal SFVhpi_SAM106-specific sequencing primers to ensure sufficient coverage.

### 2.4. Complete SFVhpi_SAM106 Genome Sequence Analysis

Gene annotation tools in CLC Genomics Workbench were used to locate open reading frames (ORFs) within coding regions of the SFVhpi_SAM106 genome. Positions of the complete 5′ and 3′ long terminal repeats (LTRs) were determined manually using previously published ape SFV genomes as a reference. Potential splice donor and acceptor positions were inferred using neural network predictions implemented in NetGene2 (http://www.cbs.dtu.dk/services/NetGene2/). N-linked glycosylation sites were predicted using the N-GlycoSite tool at the HIV LANL website (https://www.hiv.lanl.gov/content/sequence/GLYCOSITE/glycosite.html) [44]. Potential nuclear localization signals in the Tas protein were predicted using NucPred (https://nucpred.bioinfo.se/cgi-bin/single.cgi) and PSORTII (https://psort.hgc.jp/form2.html). Coiled-coil motifs were inferred using the website https://embnet.vital-it.ch/software/COILS_form.html. Nuclear export signals (NESs) were inferred using neural networks and hidden Markov models at http://www.cbs.dtu.dk/services/NetNES/.

Using Geneious v7.0.6, we extracted the five coding regions, *gag, pol, env, tas*, and *bet*, and aligned them with representative SFVs with complete genomes from four other apes, two OWMs, four NWMs, one prosimian, and one FV each from equine, bovine, and feline hosts (Table 1). We also created a concatamer of the major FV coding regions (group-specific antigen (*gag*), polymerase (*pol*), and envelope (*env*)) to enable maximally robust phylogenetic analysis. Concatamers of major coding regions of other slowly evolving cell-associated retroviruses are commonly used for evolutionary analyses [45,46,47]. Finally, since recombination was reported recently in the surface protein (SU) region of *env,* we also prepared an alignment of the complete SFV *env* sequences of species in Table 1 and those from chimpanzee, gorilla, humans, and macaques used in the analyses by Galvin et al. and Richard et al. consisting of 48 taxa [19,48] All alignments were checked for evidence of potential recombination events using first a 400-nt window and a 40-nt step, and then a 200-nt window and 20-nt step, using the Kimura-2 parameter nucleotide substitution model with gap stripping in Simplot v 3.5.1 [49]. We also checked for recombination by using the Recombination Detection Program (RDP) v4 with the default parameters [50].

We performed codon-based nucleotide alignments using MAFFT v 7.017 [51], followed by manual adjustments and gap stripping. We used the model selection algorithm in MEGA v6 [52] to determine the best fitting nucleotide substitution model, which was inferred to be the general time reversible (GTR) model with gamma (Γ) distribution and invariable sites (GTR+Γ+I). Likelihood mapping of quartet topologies implemented in IQ-TREE v1.6.8 was used to check for evidence of good phylogenetic signal in the alignment [53]. We also checked the phylogenetic signal and evidence of nucleotide substitution saturation in the alignments with the program DAMBE v7.0.35 (http://dambe.bio.uottawa.ca/DAMBE/dambe.aspx).

Phylogenies and divergence dating were simultaneously inferred using Bayesian inference with the program BEAST v.1.8.4 [54]. For BEAST analysis, we created six taxon groups, including Hominoidae (great apes), *Pan/Gorilla*, NWMs, OWMs, OWM/Hominoidae, and all simians. We used the MAFFT alignments and enforced monophyly for both simian and non-simian taxon groups in the analyses. To evaluate the potential effect of nucleotide heterogeneity sometimes observed at third codon positions of RNA viruses on the phylogeny, we also conducted phylogenetic analysis after stripping the third codon positions (cdp) from the alignment. We used an uncorrelated lognormal relaxed molecular clock, a birth-death speciation tree prior, and 100 million Markov Chain Monte Carlo (MCMC) iterations with a 10% burn-in. To more accurately estimate divergence dates, we set priors for the time to the most recent ancestor (TMRCA) dates across the FV phylogeny using normal distribution priors and nuclear DNA split estimates for NHPs and other matching non-simian placental mammals from www.timetree.org [55,56] as follows: 18.6–20.2 million years ago (mya), standard deviation (SD) 0.82 for Hominoidae; 8.44–9.06 mya, SD 0.33 for the *Pan troglodytes*/*Gorilla* split; 27.61–29.44 mya, SD 0.94 for the OWM/Hominoidae split; 74–78 mya, SD 2.0 for the *Equs caballus/Bos taurus* split; and 91–96 mya, SD 3.1 for *Boreoutheria* (placental mammals).

We used Tracer v1.6 to ensure all parameters converged with effective sampling size (ESS) values >250. Trees were logged every 10,000 generations. Two independent BEAST runs were performed to ensure convergence and reliability of the results. We used TreeAnnotator v1.8.3 to choose the maximum clade credibility tree from the posterior distribution of 10,001 sampled trees with a burn-in value of 1000 trees. The inferred trees were visualized using FigTree v1.4.2 http://tree.bio.ed.ac.uk/software/figtree/).

### 2.5. Comparison of FV tRNA Binding Motifs

tRNA primer binding site sequences for SFVhpi_SAM106 were identified using tRNAdb (http://trnadb.bioinf.uni-leipzig.de), a database that can be queried for tRNA binding motifs and outputs consensus and features of conservation for any selected set of tRNAs. FV tRNA motifs were compared to investigate potentially divergent primer binding sites.

### 2.6. Nucleotide Sequence Accession Number

The complete genome of SFVhpi_SAM106 has been deposited in GenBank with the accession number M621235.

## 3. Results

### 3.1. SFVhpi_SAM106 Genome Assembly

Assembly of 38,000 paired-end reads yielded the complete SFVhpi_SAM106 coding genome with 380× coverage. The longest contig obtained by *de novo* assembly was 11,815 nt. The read lengths ranged from 175 to 250 bp. The average read length was 203.59 bp. The sequence of the complete genome was determined by manual alignment of overlapping 5′ and 3′ LTR regions to give a final length of 13,885 nt.

### 3.2. Organization of the SFVhpi_SAM106 Genome and Comparison with Other Ape SFVs

The SFVhpi_SAM106 genome consists of all expected structural, enzymatic and auxillary gene coding regions, including *gag*, *pol*, *env*, *tas*, and *bet*, together flanked by two LTRs (Figure 1).

Gene and genome length comparisons with other ape SFVs are provided in Table 2. SFVhpi_SAM106 has the longest recorded genome among ape SFVs owing to its longer LTRs, whereas the lengths of the five coding regions are comparable in size to those from other ape SFVs.

Nucleotide and amino acid identity comparisons of the major genes and proteins, respectively, of SFVhpi_SAM106 to those of other ape FVs are provided in Table 3. Sequence analysis showed the SFVhpi_SAM106 genome was nearly equidistant from other ape SFVs, sharing approximately 65% nucleotide identity across the *gag–pol–env* region of the coding genome. The highest gene and protein identities were seen with *pol*/Pol followed by *env*/Env, *gag*/Gag, *tas*/Tas, and *bet*/Bet.

The LTR, at 2071 nt (positions 1–2071), was found to be the longest among the ape SFVs by about 300 to 800 nucleotides. We confirmed the LTR length by PCR using SFVhpi_SAM106 PCR primers and Sanger sequencing. Both the SFVhpi_SAM106_LTR-RU5 and *bet*-LTR fragments were 100% identical to the LTR obtained by NGS and were 1968 bp and 2029 bp in length after removing the primer sequences. The U3 region extends from positions 1 to 1704, which is about 300 nt longer than that of other ape SFVs; followed by the R region (positions 1705 to 1911), which is about 20 nt longer than that of other ape SFVs; and the U5 region (positions 1912 to 2071), which is about the same length as that in other ape SFVs. Three TATA box motifs were found at nucleotide positions 76, 239, and 396 upstream of the primer binding site (PBS), which is in turn 104 nt upstream of the start codon for *gag*. The poly A motif (AATAAA) is located at position 1889, the conserved 3′ and central polypurine tracts (AGGAGAGGG) are located at positions 7973 and 11,813, respectively. The first two dimerization signals (DS) were highly conserved and are located at positions 2081 and 2140, but a potential third DS was not strictly conserved and consisted of AAAAGTC instead of AAAATGG found in other SFVs [57]. There are two Ets-1 transcription factor binding domains in the 5′ LTR at positions 402 and 766. There are also three CAP (catabolite activator protein transcription motif) sites at positions 1647, 1692, and 1808. We also identified a polypurine tract (PPT) at positions 11,795, just upstream of the start of the 3′ LTR. The conserved tRNAlys PBS motif 5′-TGG CGC CCA ACG TGG GGC-3′ at positions 2074 to 2091 in SFVhpi_SAM106 is present in all available complete ape SFV genomes. The relatively conserved SFV Env/Orf-2 splice donor (AG**TTG^GTAA**TTT) and acceptor (TTTTA**AG^A**TAAT) sites are located at positions 10,292 and 10,411, respectively, and were predicted by NetGene2. The nucleotide identity of the SFVhpi_SAM106 LTR to other ape SFV LTRs ranged from 30% to 45%, with the closest identity to LTRs from chimpanzee SFVs.

The internal promoter (IP) was identified by comparison with other FVs and is located at *env* nucleotide positions 10,142 to 10,198 (5′-CAA GAG AA **CATAAA** AGA TCA AAT CGA GAG AGC AAC CGC AGA GC-3′) (Figure 1). However, unlike the TATAAA box consensus motif of exogenous FV IPs, the SFVhpi_SAM106 TATA promoter box is more similar to that of two endogenous FVs (sloth and coelacanth) in that it starts with a cytosine (highlighted in the IP sequence in bold and italics) instead of a thymine [58]. A potential CAP site was identified 48 nucleotides downstream of the IP. Fifteen N-linked glycosylation sites were identified, of which one is in the LP region, 11 are in SU, and three are in TM. Ten of these are NXT variants and five are NXS. In comparison, SFVppy has 13 N-linked glycosylation sites of which one is in LP, nine are in SU, and three are in TM. Nine of these are NXT variants and four are NXS.

The number, position, and composition of potential *tas*-response elements (TREs) varies among SFVs and requires in vivo experiments for confirmation [58]. By comparison with other SFVs, we identified a potential TRE upstream of the IP at position 10,021 (CTTAAAGGCAGAAGAGAAA). TREs in the LTR region could not be readily identified.

We observed that *gag* (1974 nt; positions 2178–4151), *pol* (3420 nt; positions 4102–7521), *env* (2966 nt; positions 7481–10,446), and *tas* (897 nt; positions 10,420–11,316) ORF lengths are similar to other ape SFVs (Table 2 and Table 3). To identify the potential splice/acceptor sites for the *bet* gene, we used an online neural network prediction tool. Although we found evidence of a splice donor site in SFVhpi_SAM106 at positions 10,693 to 10,702 (5′-GAGGAATGA^TAAGTTAAT-3′) and a splice acceptor site at positions 10,991 to 11,010 (5′-CTCCTATTAG^GTACACTGGG-3′) indicating possible bet splicing, support was not strong (0.54 and 0.30, respectively). We also found a splice acceptor site within the bet ORF (positions 11,375–11,395, 5′-AATTCTCAG^ATGATGAGGAT-3′) with strong support (0.96), which could potentially give rise to an alternate *tas-bet* fusion transcript 1065 nucleotides and 355 amino acids in length.

Notable motifs identified *in silico* in Gag include the cytoplasmic retention and targeting signal (CTRS) (GEWGFGD**R**YNVVQIVLQD) located at aa positions 39 to 56 with the highly conserved arginine (R) at position 46 essential for intracytoplasmic particle formation. The YXXL motif involved in particle assembly was highly conserved at positions 77 to 80. The P3-cleavage site (RSFN/TVSQ) is located at aa positions 624 to 631 in the carboxyl terminus. Analysis of the Gag sequence did not identify conserved P(T/S)AP late domain motifs in the center of the protein, but we did find two PPAP motifs at aa positions 269 to 272 and 296 to 299. These motifs are similar in number and location to those in SFVpve and SFVggo, but not SFVppy, in which the single PSAP motif is located near the end of Gag. We also identified an assembly domain (YEMLGL) at aa positions 462 to 467. Examination of Gag for glycine-arginine (GR) rich boxes involved in viral replication identified four potential GR-boxes at aa positions 485 to 509 (GGRGRGRNNRNAASGNTQGGNQRQSR), 515 to 534 (GRQSQGGRGRGSNNNTNSRQ), 538 to 562 (QNSSGYNLRPRTYNQRYGGGQGRR), and 595 to 612 (RGDQPRRSGAGRGQGGNR) compared to the three such motifs found in other SFVs. The highly conserved chromatin binding motif was located in the third GR at AA positions 542 to 548 (underlined above), which for other FVs is in GRII [59]. However, a nuclear localization signal (NLS) present in GRII of SFVpsc was not identified in the SFVhpi_SAM106 GRIII by using PSORTII and NucPred [59]. An NES at the N-terminus of Gag has been shown to be critical for the late stages of virus replication of SFVpsc (formerly HFV) and is partially conserved in SFVmcy and SFVcae [60]. Comparison with other SFVs identified a potential NES at aa positions 91 to 108 (LAFNGIGPAEGALRFGPL); however, NetNES predicted in the SFVhpi_SAM106 Gag a leucine-rich NES around aa positions 8 to 20 (LDVQELVLLMQDL), which is relatively conserved in other ape SFV Gag proteins. NetNes correctly predicted the reported NES in SFVpsc, SFVpve, and SFVggo, but not in SFVppy, SFVmcy, SFVcae, SFVocr, SFVcja, SFVaxx, SFVssc, SFVsxa, BFVbta, and EFVeca. An NES similar to that in SFVhpi_SAM106 was predicted for FFVfca.

Within the Pro-Pol polypeptide, the highly conserved protease catalytic center (DTGA) and reverse transcriptase (RT) catalytic site (YVDD) were located at aa positions 24 to 26 and 312 to 315, respectively. The DSF motif required for RNAse H activity located at positions 670 to 672 was also conserved. The viral protease cleavage site for the integrase (IN) protein (YTVN/NIQN) was partially conserved (YVNN/XNXX) and located at aa positions 749 to 756, potentially coding for a 388 aa IN peptide. The IN catalytic center (DD35E) was found at D^898^D^936^E^972^ and the zinc-finger motif at H^813^H^817^C^847^C^850^. Interestingly, we also identified a highly conserved TATA box motif at positions 4840 to 4846 in *pol* present in the RT region of all FVs with the consensus sequence (T(A/T)(T/C)AA(A/G) (Figure 2). In SFVhpi_SAM106, this TATAAA motif is 80 nucleotides upstream of the RT active site.

For the 989-aa Env protein, the highly conserved WXXW motif required for Gag interaction and budding is located at aa positions 10 to 13 (WLIW). The cellular furin protease cleavage sites, which cleave the N-terminal leader peptide (LP) from the SU and the SU from the transmembrane (TM) protein, are located at aa positions 125 to 131 (RLAR/RSLR) and 570 to 577 (RKRA/TSSN) to generate three potential Env proteins of lengths 127 (LP), 446 (SU), and 416 (TM), respectively. The highly conserved hydrophobic WXXW motif in the LP required for Env incorporation and particle release is located at aa positions 10 to 13. The membrane-spanning domain located at aa positions 945 to 980 (AKGIFGTAFSLVAYVKPILIGIGVIILLVVIFKIIS) is partially conserved. The consensus KKK endoplasmic reticulum retrieval signal located at aa positions 694 to 696 of the TM (or positions 985 to 987 of Env) is partially conserved and has the sequence KAK, whereas SFVppy, SFVaxx, SFVggo_BAK74, and SFVssc motifs have the sequence KRK. A putative receptor-binding domain (RBD) is located in SU at aa positions 227 to 552 and the relatively conserved fusion peptide (LRSMGYALTGGIQTVSQI) is located in the TM at aa positions 581 to 598 of Env [61].

The ape SFV Tas proteins are highly divergent (Table 3) and hence identification of the poorly defined acidic activation and DNA binding domains was not possible. Tas is a nuclear protein involved in transcriptional transactivation with a partially conserved NLS at the 3′ end of the protein. In SFVhpi_SAM106, the NLS GTGRKRRTN is located at aa positions 216 to 224 with a strong NucPred score of 0.87, whereas PSORT II identified RKRR in this motif as an NLS, but with a low score (−0.16). Neither prediction program found a bipartite NLS in SFVhpi_SAM106 or other ape SFV Tas proteins. PSORT II predicted with high reliability (94.1) that the SFVhpi_SAM106 Tas was a nuclear protein, similar to the Tas protein of SFVppy (reliability score = 94.1), but more likely than those from SFVggo and SFVpve/psc (reliability score = 89). The PSORT II predictor uses a heuristic algorithm, including neural networks, to identify nuclear proteins are rich in basic residues. The SFVhpi_SAM106 Tas includes 41 basic amino acids (R = 25, K = 12, H = 4), or about 13.7% of the protein, compared to 12.2% (34; R = 15, K = 15, H = 4) for SFVppy, 14.1% (42; R = 14, K = 20, H = 8) for SFVggo, 16.6% (50; R = 16, K = 22, H = 12) for SFVpve, and 15.3% (46; R = 18, K = 17, H = 11) for SFVpsc. A leucine-rich NES was predicted to be at aa positions 97 to 107 (LICERLILLAL).

Although the *bet* splice acceptor site described above was not strong, the SFVhpi_SAM106 Bet protein length of 483 aa was similar to that of other ape SFVs (SFVpve = 490 aa, SFVpsc = 482 aa, SFVggo = 481 aa, SFVppy = 464). Comparison with these other ape SFV Bet proteins identified a potential integrin-binding motif (K/RGD) at aa positions 306 to 308 that was partially conserved (KGT), with SFVpve, SFVggo, and SFVppy having a KGD motif and SFVpsc having an RGD motif. The tripeptide RGD domain has been shown to be required for cell membrane binding, so it will be important to examine the functionality of the D > T mutation at the third aa position in the SFVhpi_SAM106 Bet. One study has proposed the SFVpve Bet is secreted in both the cytoplasm and nucleus of infected cells and contains a bipartite NLS in the C-terminus [62]. PSORT II does detect the bipartite NLS RKIRTLTEMTQDEIRKR at aa positions 463 to 479 of SFVpsc Bet, but with a cytoplasmic protein reliability prediction of 70.6% rather than a nuclear one. NucPred did not predict a NLS in the SFVpsc Bet. Neither NucPred nor PSORT II identified any putative NLS in the SFVhpi_SAM106, SFVpve, SFVggo, and SFVppy Bet proteins.

### 3.3. Absence of Evidence of Genetic Recombination in the SFVhpi_SAM106 Genome

Simplot analyses on the *gag-pol-env* concatamer alignment did not show any evidence of genetic recombination at a threshold of 70% of permuted trees, a cutoff commonly used for the analysis of other retroviruses (data not shown). These results were consistent across two window and step sizes in the analysis. We also found no significant evidence of recombination using the recombination detection program RDP4 using eight methods (RDP, GENCONV, Chimaera, MaxChi, BootScan, SiScan, 3Seq, and LARD). We did not find any evidence of recombination of the SFVhpi_SAM106 env using the 48 taxa dataset by phylogenetic and RDP analysis, but we did confirm the two different SU RBD SFV clades as previously reported (Appendix A) [19,48]. Phylogenetic analysis of the two alignments encompassing the complete *env* and the region without the RBD in SU showed the typical co-evolutionary history of FV with SFVhpi_SAM106 ancestral to SFVppy, SFV OWMs, and then chimpanzees and gorillas with strong support. In the analysis of only the BD region of SU, SFVhpi_SAM106 was ancestral to the chimpanzee and gorilla SFVs, but with no support. In addition, SFVppy clustered FFVfca with good support between the NWM SFV and the other OWMA SFVs. Combined, these results suggest an absence of recombination in SFVhpi_SAM106 and that genetic recombination did not affect our phylogenetic results when using the *gag-pol-env* concatamer.

### 3.4. Evolutionary Relationships and Divergence Dating of SFVhpi_SAM106 and Other FVs

Likelihood mapping of the 15 taxa 7,412 position *gag-pol-env* concatamer alignment showed an equal distribution of the majority of possible quartets across the tree of which 98.4% were fully resolved, i.e., tree-like, with only 1.68% of unresolved quartets. These results suggest an overall dataset with very good phylogenetic signal and very little “noise” and hence suitable for phylogenetic reconstruction. Excellent phylogenetic signal was also found in both *gag-pol-env* alignments with scores >99.3 using DAMBE. Little nucleotide substitution saturation was found in the alignments using the method of Xia in DAMBE [63]. Together, our results indicate the alignments were satisfactory for phylogenetic analysis.

Phylogenetic trees generated using Bayesian inference of the *gag-pol-env* concatamer showed that FV sequences from a broad range of genetically diverse NHPs and non-simians formed monophyletic lineages and distinct clusters that mirrored host taxonomic relationships (Figure 3). SFVhpi_SAM106 clustered with other ape SFVs with strong posterior probability (PP > 1) support (Figure 3). The FV phylogeny was similar to that seen in our previous study, where SFVhpi_SAM106 is a sister taxa of but ancestral to the great ape SFVs, mirroring the phylogeny of the host mitochondrial and nuclear sequences in which the lesser apes are an outgroup to the great apes [6,42]. An identical phylogeny was obtained with the *gag-pol-env* first and second cdp alignment, indicating the absence of substitution saturation at the third cdp in the analysis of the unstripped alignment (Figure 3). As expected, the representative prosimian FV sequence from a galago (SFVocr) was ancestral to all other SFVs. The three non-simian FVs from equine, feline, and bovine formed a clade separate from the SFVs, with BFV and EFV clustering together with strong support (PP = 1). All BEAST analyses had standard deviation values of the uncorrelated lognormal relaxed clock (ucld.stdev) greater than zero but less than one, indicating that variation in substitution rates across branches was not consistent with a strict molecular clock but also was not so great as to bias the analyses (Table 4). The absence of site-to-site variation was also supported by alpha parameters of the gamma distribution above 1.0 (Table 4).

Bayesian dating analyses showed that for the *gag-pol-env* concatamer alignment, the TMRCA for SFVhpi_SAM106 and the Hominoidea was 20.69 mya with a 95% highest posterior density (HPD) interval of 19.13 to 22.19 mya (Table 4). This divergence date occurs during the Miocene epoch (Figure 3). The TMRCA for the SFV OWM/ape split (Crown Catarrhini) was 27.28 mya with a 95% HPD interval of 27.28 to 30.09 mya. The SFVpsc, SFVpve/SFVggo split (Crown Homininae) had a TMRCA of 9.18 mya with a 95% HPD interval of 8.54 to 9.8 mya. For the OWM SFV (Crown Cercopithecinae), a TMRCA of 18.05 mya was inferred with a 95% HPD interval of 11.29 to 24.7 mya. In contrast, TRMCAs for the non-simian FVs were much older, estimated to be 92.4 mya (95% HPD interval of 85.03–98.69 mya). The TMRCA for the Boreoutherian (placental mammals) at the root of the FV phylogeny was estimated to be 95.46 mya (95%HPD interval of 90.08 to 101.19 mya) during the Upper epoch and Mesozoic era. Similar TMRCAs were inferred for the 12 cdp alignment (Table 4). Our inferred FV and host divergence dates were similar to those inferred by others using different methods, supporting the robustness of our dating methods (Table 4). The inferred mean substitution rates ranged from 5.05 × 10^−9^ to 9.82 × 10^−9^ nucleotides/site/year across the *gag-pol-env* coding region (Table 4).

## 4. Discussion

Using a metagenomics approach, we obtained the first complete genome of an SFV from a lesser ape, the pileated gibbon (*Hylobates pileatus*). Next generation sequencing is a powerful molecular method and has been used to obtain complete genomes of other SFV isolates recently [13,67,68]. We characterized the SFVhpi_SAM106 genome by detailed sequence analysis and provided a deeper understanding of the evolutionary history of FVs, especially within the Hominoidea. Most of the important genomic structures and functional domains were conserved in SFVhpi_SAM106 with some exceptions. While the organization of the SFVhpi_SAM106 genome is similar to that of other FVs, it has several unique features, including an LTR at ~2 kb that is 1.2 to 1.6 times longer than that in other FVs, attributable mostly to longer U3 and R regions. Given that the 5′ LTR U3 and R regions of FVs contain transcriptional start control elements, including the *tas* response elements (TREs) and cellular transcription factors, it will be important to conduct transcription-mapping experiments to determine the functional significance of the longer LTRs of SFVhpi_SAM106 for replication and regulation. Since the number and location of FV TREs are not conserved, we were unable to locate the SFVhpi_SAM106 TREs within the LTRs or *env* sequences *in silico*. Others have shown that tissue culture of chimpanzee SFV in diploid human fibroblasts isolated from an infected human selects for isolates with substantial nonrandom nucleotide deletions in the U3 region can increase viral replication [69]. The U3 deletion variant was also present as the majority variant in the infected human but was absent from naturally-infected chimpanzees [69]. However, SFVhpi_SAM106 was isolated from infected PBMCs using canine thymocyte cells [42], which have not been shown to impact SFV LTR length or functionality in this dog cell line or in other non-human cells, including SFVggo and SFVcpz isolates from zoonotically-infected humans grown in baby hamster kidney cells (BHK-21) [57]. Comparisons of SFVhpi_SAM106 LTRs directly from PBMCs from the pileated gibbon and those obtained by culture may be needed to further evaluate this unusually long LTR.

Additional differences from other FV prototypes included four instead of three potential GR boxes in Gag and a TATA box promoter motif in the IP in *env*, which is more similar to those of two endogenous FVs from the sloth and coelacanth by starting with a cytosine instead of a thymine present in other FVs [58]. Since the GR and TATA box motifs are important for viral replication, studies are needed to determine the effect of these genetic differences on SFVhpi_SAM106 replication and growth. Some subtypes of human and simian immunodeficiency viruses (HIVs and SIVs) have the CATAAA TATA box sequence in their LTRs, which have not been shown to decrease LTR promoter activity by the HIV-1 transcription activator (Tat) protein in vitro [70]. Examination of the SFVhpi_SAM106 genome for additional TATA promoters identified a highly conserved TATA box upstream of the RT active site promoter in the FV genome and its possible effect on viral replication is needed.

FVs have been shown to have co-evolved with their hosts for millions of years as vertebrates diversified in the Paleozoic Era [5,6,64]. Our results expand knowledge of the evolutionary history of FVs and show for the first time that SFVhpi_SAM106 from the lesser ape, *Hylobates pileatus,* follows this same ancient co-speciation trajectory. The phylogenetic position of SFVhpi_SAM106 mirrors that of the gibbon host, with a divergence date of about 21 mya (95% HPD 19.13–22.19 mya) during the Miocene Epoch of the Cenozoic Era. The FV divergence dates inferred with our methods are strongly congruent with those of Aiewsakun and Katzourakis, who used Bayesian phylogeny and a time-dependent rate power-law decay function that is independent of archaeological calibrators [64]. Our Hylobatidae/Hominidae divergence dates are also consistent with those of Matsudaira et al. (20.3 mya, 95% CI 17.5–23.6 mya) and Chan et al. (19.25 mya, 95% HPD 15.54–22.99), who examined ape complete mitochondrial sequences [71,72]. The Hylobatidae/Hominidae divergence dates are slightly older than those of reported by Thinh et al. (16.26, 95% HPD 14.69–18.16 mya), who used only complete cytochrome B mitochondrial sequences [38]. In addition, our inferred SFV divergence dates are also highly consistent with those reported for the evolutionary histories of NHPs and other placental mammals examined in our study and had very close divergence dates with overlapping confidence intervals (Table 4). Together, these results show that our use of multiple host divergence calibration points provided a robust inference of FV evolutionary histories. As with all evolutionary studies, the inferred divergence dates represent minimum ages that are younger than true ages, because the fossil record is incomplete.

Given the strong evidence of FVs co-diverging with their hosts, both FVs and their mammalian hosts [73] should have very similar evolutionary rates. Indeed, FVs have extremely low rates of evolution, similar to rates observed for mitochondrial protein-coding genomes of about 1 × 10^−8^ substitutions/site/year [4,6,11]. However, our estimates were a log lower and are more similar to those of the mammalian neutral substitution rate for nuclear genes and for endogenous retroviruses of 1 × 10^−9^ substitutions/site/year [74,75]. Our lower inferred FV nucleotide substitution rates may reflect the use of older calibration dates for the *Equs caballus/Bos Taurus* split and for Boreoutheria in the phylogenetic analyses.

Genetic recombination did not appear to affect our phylogenetic results. Phylogenetic analysis of only the FV *env* region with additional SFV taxa from the gorilla, chimpanzee, and macaque with evidence of a variant RBD from a potential recombination event showed that SFVhpi_SAM106 was ancestral to SFVppy and then all other OWMA SFVs instead of ancestral to only the apes as we found for the *gag-pol-env* concatamer alignments. Similar results were inferred even after removal of the putative recombination region in the RBD of SU, whereas in the RBD-only alignment, placement of the SFVhpi_SAM106 and SFVppy taxa were not resolved. Similar genetic relationships were observed by Richard et al. in SFVppy when examining complete gorilla and chimpanzee *env* sequences [19]. One exception in their analysis of the variant RBD region is that SFVppy clustered ancestral to chimpanzee and gorilla clade 1 instead of with FFVfca as in our analysis since non-simian FVs were not included in their analysis [19,48]. Such nonconforming FV co-evolutionary phylogenetic relationships may reflect the phenomenon called long branch attraction (LBA), which can occur when highly divergent lineages are included in the phylogenetic analysis [73]. LBA can potentially be resolved by the addition of additional sequence information [73], which we have done by using the *gag-pol-env* concatamer, or by inclusion of more SFV sequences from gibbons and orangutans when they become available.

As with other NHPs, gibbons are threatened by habitat encroachment for forest clearance, road construction, and changes in agriculture resources. Gibbons are also hunted for food, medicine, or for exportation for the pet trade (see also the International Union for Conservation of Nature’s Red List at www.iucnredlist.org) [40,76]. Gibbons are also common members of zoological exhibits. All of these activities can lead to human exposures to gibbons and their microbial communities, including SFVs, which have crossed into humans from many NHP species [12]. Until now, only two short SFV integrase sequences were available to inform the design of sensitive PCR assays for the detection of human infection [42]. More sensitive molecular assays are needed to detect the low copies of integrated genomes typically found in SFV infection. The small number of complete SFV genomes from all available NHPs has also limited the design of generic SFV PCR primers. These limitations may help explain the inability to detect SFV sequences in seropositive pileated gibbons from Cambodia using generic PCR assays [28]. The availability of the complete SFVhpi_SAM106 genome from our study will facilitate the design of more sensitive and specific PCR assays for the detection of gibbon SFVs and fills an important knowledge gap in the SFV database, facilitating future studies of FV infection, transmission, and the evolutionary history of FVs.

## 5. Conclusions

By using a metagenomics approach, we obtained the complete SFVhpi_SAM106 genome from a pileated gibbon (*Hylobates pileatus*). Bayesian analysis showed that SFVhpi_SAM106 is ancestral to other ape SFVs with a divergence date of ~20.6 million years ago, reflecting ancient co-evolution of the host and virus. Our molecular analysis also showed that the SFVhpi_SAM106 genome has the longest genome (13,885 nt) of all SFVs with complete genomes available, due to a much longer LTR (2071 bp). The complete sequence of the SFVhpi_SAM106 genome will provide invaluable information for further understanding the epidemiology and evolutionary history of SFVs.

## Figures and Tables

**Figure 1 viruses-11-00605-f001:**
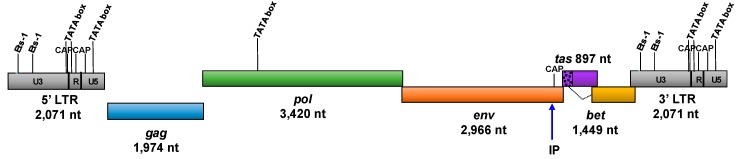
Genomic structure of SFVhpi_SAM106. Ets-1, ETS proto-oncogene 1 transcription factor motif; CAP, catabolite activator protein transcription motif; TATA box, promoter region motif; LTR, long terminal repeat; IP, internal promoter; *gag*, group-specific antigen; *pol*, polymerase; *env*, envelope; *tas*, transactivator gene; *bet*, between *env* and *tas* genes; U3, unique 3′ region of the LTR; R, repeat region of the LTR; U5, unique 5′ region of the LTR. The Bet protein is translated from a spliced RNA and residues from the 5′ part of *tas* indicated by the speckled region and dotted line.

**Figure 2 viruses-11-00605-f002:**
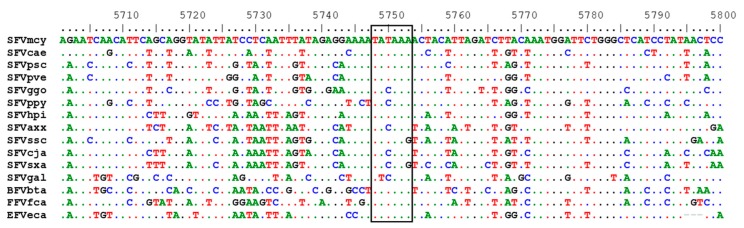
Conserved TATA box in alignment of foamy virus polymerase sequences. Dots represent conserved nucleotides relative to the first sequence. Nucleotide positions are after alignment using MAFFT. Old World apes (OWA): SFVpve, *Pan troglodytes verus* (chimpanzee), GenBank accession number U04327; SFVpsc, *Pan troglodytes schweinfurthii* (chimpanzee), Y07725; SFVppy, *Pongo pygmaeus* (orangutan), AJ544579; SFVggo, *Gorilla gorilla* (gorilla), NC_039029; SFVhpi_SAM106, *Hylobates pileatus*, (pileated gibbon) M621235. Old World monkeys (OWMs): SFVcae, *Cercopithecus aethiops* (African green monkey), M74895; SFVmcy, *Macaca cyclopsis* (macaque), X54482. New World monkeys (NWM): SFVcja, *Callithrix jacchus* (common marmoset), GU356395; SFVsxa, *Sapajus xanthosternos* (capuchin), KP143760; SFVaxx, *Ateles* species (spider monkey), EU010385; SFVssc, *Saimiri sciureus* (squirrel monkey), GU356394. Prosimian (Pro): SFVocr, *Otolemur crassicaudatus* (brown greater galago), KM233624. Non-simian mammals (NSM): EFVeca, *Equus caballus* (equine), AF201902; BFVbta, *Bos taurus* (bovine), U94514; and FFVfca, *Felis catus* (feline), Y08851.

**Figure 3 viruses-11-00605-f003:**
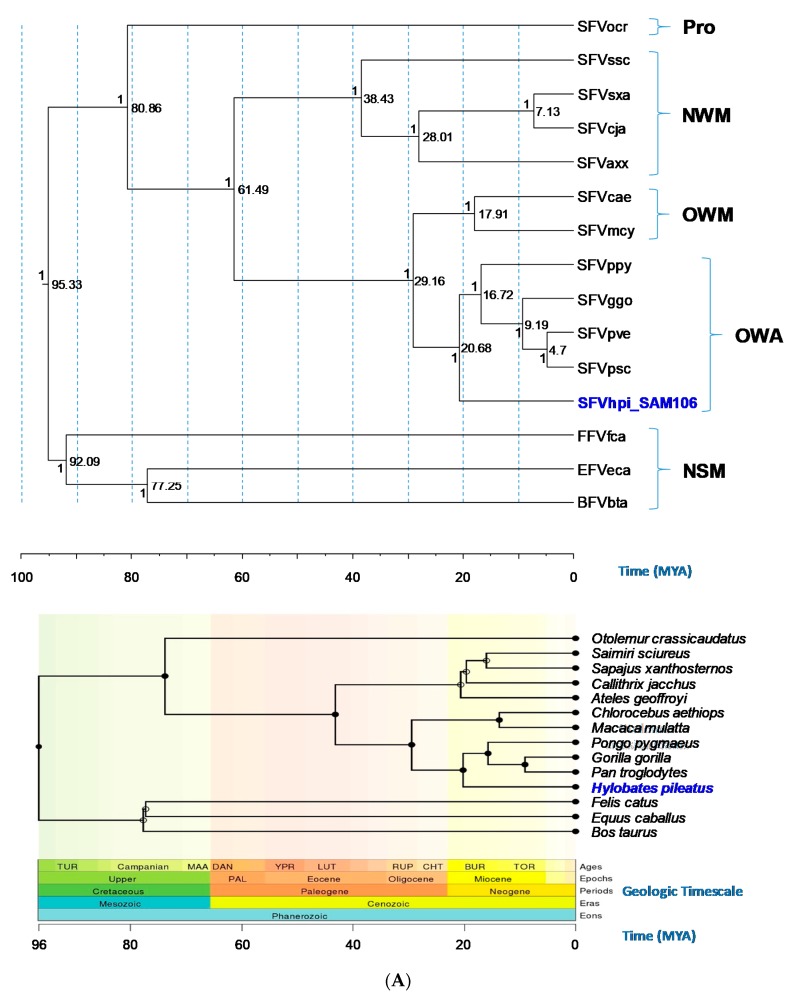
Evolutionary relationships and divergence dates of foamy viruses (FVs) and their mammalian hosts. (**A**) FV phylogeny inferred using *gag-pol-env* concatamer (~7.4 kb). (**B**) FV phylogeny inferred using the first and second coding positions of the *gag-pol-env* concatamer (~4.9-kb). Phylogeny inferred using BEAST and mammalian host phylogeny inferred at Timetree.org as well as the corresponding geologic timescale. Old World apes (OWAs): SFVpve, *Pan troglodytes verus* (chimpanzee), GenBank accession number U04327; SFVpsc, *Pan troglodytes schweinfurthii* (chimpanzee), Y07725; SFVppy, *Pongo pygmaeus* (orangutan), AJ544579; SFVggo, *Gorilla gorilla* (gorilla), NC_039029; SFVhpi_SAM106, *Hylobates pileatus*, (pileated gibbon) M621235. Old World monkeys (OWMs): SFVcae, *Cercopithecus aethiops* (African green monkey), M74895; SFVmcy, *Macaca cyclopsis* (macaque), X54482. New World monkeys (NWM): SFVcja, *Callithrix jacchus* (common marmoset), GU356395; SFVsxa, *Sapajus xanthosternos* (capuchin), KP143760; SFVaxx, *Ateles* species (spider monkey), EU010385; SFVssc, *Saimiri sciureus* (squirrel monkey), GU356394. Prosimian (Pro): SFVocr, *Otolemur crassicaudatus* (brown greater galago), KM233624. Non-simian mammals (NSMs): EFVeca, *Equus caballus* (equine), AF201902; BFVbta, *Bos taurus* (bovine), U94514; and FFVfca, *Felis catus* (feline), Y08851). Posterior probabilities (on the branch left of the node) and time to most recent ancestors in millions of years (right of node) are provided at each node of the FV phylogeny. Solid circles in the Timetree.org mammalian phylogeny indicate nodes that map directly to the NCBI taxonomy and open circles indicate nodes that were created during the polytomy resolution process. TOR, Tortonian; BUR, Burdigalian; CHT, Chattian; RUP, Rupelian; LUT, Lutetian; YPR, Ypresian; DAN, Danian; MAA, Maastrichtian; TUR, Turonian ages. PAL, Paleogene epoch.

**Table 1 viruses-11-00605-t001:** Foamy virus complete genomes analyzed.

Foamy Virus	Mammalian Host	Host Scientific Name	Family	GenBank Accession Number
SFVhpi_SAM106	Pileated gibbon	*Hylobates pileatus*	*Hylobatidae*	M621235
SFVpve	Western chimpanzee	*Pan troglodytes verus*	*Hominidae*	U04327
SFVpsc	Eastern chimpanzee	*Pan troglodytes schweinfurthii*	*Hominidae*	Y07725
SFVppy	Bornean orangutan	*Pongo pygmaeus*	*Hominidae*	AJ544579
SFVggo	Lowland gorilla	*Gorilla gorilla gorilla*	*Hominidae*	NC_039029
SFVcae	African green monkey	*Cercopithecus aethiops*	*Cercopithecidae*	M74895
SFVmcy	Formosan rock macaque	*Macaca cyclopsis*	*Cercopithecidae*	X54482
SFVcja	Common marmoset	*Callithrix jacchus*	*Callitrichinae*	GU356395
SFVsxa	Yellow-breasted capuchin	*Sapajus xanthosternos*	*Cebinae*	KP143760
SFVaxx	Spider monkey	*Ateles* species	*Atelinae*	EU010385
SFVssc	Common squirrel monkey	*Saimiri sciureus*	*Saimirinae*	GU356394
SFVocr	Brown greater galago	*Otolemur crassicaudatus*	*Galagidae*	KM233624
EFVeca	Horse	*Equus caballus*	*Equidae*	AF201902
BFVbta	Cow	*Bos taurus*	*Bovidae*	U94514
FFVfca	Cat	*Felis catus*	*Felidae*	Y08851

**Table 2 viruses-11-00605-t002:** Ape simian foamy virus (SFV) gene and genome nucleotide length comparison.

Virus ^1^	LTR	*gag*	*pol*	*env*	*tas*	*bet*	Genome
SFVhpi_SAM106	2071	1974	3420	2966	897	1449	13,885
SFVpve	1760	1959	3438	2964	900	1470	13,246
SFVpsc	1767	1944	3431	2964	900	1446	13,242
SFVggo	1283	1964	3154	2963	897	1443	12,258
SFVppy	1621	1987	3236	2965	834	1392	12,823

^1^. SFVhpi_SAM106, pileated gibbon (M621235); SFVpve, chimpanzee (U04327); SFVpsc, chimpanzee (Y07725); SFVppy, orangutan (AJ544579); SFVggo, gorilla (JQ867465); LTR, long terminal repeat; *gag*, group-specific antigen; *pol*, polymerase; *env*, envelope; *tas*, transactivator gene; *bet*, between *env* and *tas* genes.

**Table 3 viruses-11-00605-t003:** Percent nucleotide and amino acid identity comparisons of SFVhpi_SAM106 compared to SFVs of other ape hosts.

Virus ^1^	*gag*/Gag	*pol*/Pol	*env*/Env	*tas*/Tas	*bet*/Bet	Concatamer ^2^
SFVpve	49.2/38.9	74.3/75.9	67.0/66.2	49.2/31.3	49.2/27.9	65.6/63.4
SFVpsc	49.6/38.3	73.4/75.2	67.3/67.1	49.7/38.5	50.3/29.6	65.7/63.5
SFVggo	48.0/40.2	74.1/76.6	67.8/67.5	48.3/29.7	48.0/30.7	65.7/64.7
SFVppy	48.7/40.6	73.2/76.1	65.8/63.8	48.0/32.7	41.3/22.2	64.7/63.3

^1^. SFVhpi_SAM106, pileated gibbon; SFVpve, chimpanzee (U04327); SFVpsc, chimpanzee, (Y07725); SFVppy, orangutan (AJ544579); SFVggo, gorilla (JQ867465). *gag*, group-specific antigen; *pol*, polymerase; *env*, envelope; *tas*, transactivator protein; Bet, between *env* and *tas* protein. ^2^. Concatenation of *gag*/Gag, *pol*/Pol, and *env*/Env nucleotide/proteins.

**Table 4 viruses-11-00605-t004:** Time to most recent common ancestor (TMRCA) estimates in millions of years ago and nucleotide substitution rates for foamy viruses (FVs) ^1^.

FV ^2^, Host or Analysis Parameter	*FV gag-pol-env*	*FV gag-pol-env (12cdp)* ^3^	FV Pol ^4^	Host ^5^
SFVhpi_SAM106/SFVgreat ape split (Crown Hominoidea)	20.69 (19.13–22.19)	20.83 (19.32–22.45)	N/A ^6^	22.32 (20.54–23.85
SFV OWM/ape split (Crown Catarrhini)	29.13 (27.28–30.09)	29.07 (27.37–30.96)	31.60 (25.36–37.85)	32.12 (29.44–33.82)
SFVpve, SFVpsc/SFVggo split (Crown Homininae)	9.18 (8.54–9.8)	9.16 (8.53–9.8)	8.29 (6.52–10.05)	10.63 (10.02–11.68)
SFV OWM (Crown Cercopithecinae)	18.05 (11.29–24.77)	16.44 (10.18–22.81)	N/A	14.09 (12.24–15.82)
Prosimian SFV (Crown Primates)	82.53 (64.77–96.82)	80.85 (65.25–95.16)	86.92 (76.34–97.22)	74.11 (68.17–81.20)
Non-simian FV (horse/cat split)	92.4 (85.03–98.69)	93.05 (86.2–99.12)	N/A	80.6 (63.0–115.4)
BFV/EFV split (cow/horse split)	77.2 (73.26–80.85)	76.89 (73.11–80.61)	N/A	105.7 (74.1–130.8)
Root FV placental mammals (Crown Boreotheria)	95.46 (90.08–101.19)	95.86 (90.47–101.36)	98.59 (95.94–100.78)	120.5 (97.8–136.6)
Mean rate ^7^	8.68 × 10^−9^ (7.88 × 10^−9^–9.48 × 10^−9^)	5.05 × 10^−9^ (4.62 × 10^−9^–5.51 × 10^−9^)	N/A	N/A^9^
ucld.stdev ^8^	0.5524 (0.4084, 0.7252)	0.435 (0.2931–0.5911)	N/A	N/A
α-parameter (Γ-distribution) ^9^	1.079 (0.9704, 1.1858)	1.064 (0.8954, 1.2586)	N/A	N/A
ESS ^10^	>258	>1063	N/A	N/A

^1^. Median TMRCAs; 95% high posterior density values are shown in parentheses. ^2^. SFVhpi_SAM106, SFV from *Hylobates pileatus*; OWM, old world monkeys; SFVpve, SFV from *Pan troglodytes verus*; SFVpsc, SFV from *Pan troglodytes schweinfurthii*; SFVggo, SFV from *Gorilla gorilla*, non-simian foamy viruses include bovine foamy virus (BFV), equine foamy virus (EFV), and feline foamy virus (FFV). ^3^. 12 cdp, first and second codon positions. Length is 4.942 kb versus 7.412 kb for the complete coding concatamer. ^4^. FV Pol amino acid divergence dates were inferred with Bayesian phylogenetic inference and a time-dependent rate phenomenon, which is a power-law decay function [64]. ^5^. Primate and non-primate divergence dates from Pozzi et al. (complete mitochondrial genomes) and dos Reis et al. (nuclear genomes and mitochondrial genomes) [65,66]. ^6^. N/A, not available. ^7^. Mean evolutionary rate in nucleotide substitutions/site/year. ^8^. Standard deviation (stdev) of the uncorrelated lognormal (ucld) relaxed clock, BEAST analysis parameter indicating the amount of variation in the substitution rate across branches. Values close to zero indicate little substitution rate variation and the presence of a molecular clock, whereas values >1 indicate substantial rate heterogeneity amongst lineages. ^9^. Shape parameter of the gamma (Γ) distribution of rate heterogeneity among sites. ^10^. ESS, effective sampling size values for all BEAST parameters.

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
