# Peer review of "Molecular Analysis of the Complete Genome of a Simian Foamy Virus Infecting Hylobates pileatus (pileated gibbon) Reveals Ancient Co-Evolution with Lesser Apes"

_viruses, 2019, doi:10.3390/v11070605_

Round 1
Reviewer 1 Report
Shankar et al. obtained the full length genetic sequence from an SFV strain isolated from pileated gibbon. They compared this sequence to those of other SFV species, analyzed the presence of functional motifs in genes and proteins and studied the evolutionary history of this strain. This thorough presentation of the first SFVhpi sequence represents significant knowledge for further basic virology and SFV diagnosis. The paper should be enhanced by a better description of recombination analysis, a study of env gene variability, rigorous checking of functional motifs, inclusion of recently obtained SFV sequences, adequate citation of the bibliography, and use of a more fluent writing style.
Broad comments
Line 35: “the functional SFV genome is double stranded DNA rather than single stranded-DNA”. This is an overstatement, as controversial data have been published. The two cited reviews state that DNA genome is functional but do not mention data on the relative infectivity of DNA and RNA genomes.
Lines 34 and 37: two redundant sentences are separated by a third one. Please edit this paragraph and the whole manuscript for fluency.
Lines 41-43: For the description of SFV infection in NHPs, the authors cite only their papers, but neither a review, nor major papers from other teams. Lines 43-51: For the description of SFV infection in humans, the authors cite their research papers and reviews, but no research papers from other teams. In addition, Ref 22 is erroneous as it described SFV infection in macaques and not in humans. To avoid skewing towards self-citation, the authors have to cite the most recent review(s) and research papers from all teams if published after the review.
Lines 50-53: Please cite more recent reviews for the description of in vitro cytopathic effect and of SFV infection in NHPs. The authors should add the two papers that search for medical consequences in humans, published by Boneva et al. in 2007 and Buseyne et al. in 2018.
Lines 68-75: The authors provide the list of isolates with full length sequences provided in ref 28. Additional complete sequences are available in Genbank, including SFV from Papio Anubis, and from Brachyteles arachnoides. These should be used in the analysis.
Lines 100-102: The treatment of PBMCs need to be described. Where the cells fresh? Frozen? Stimulated with a mitogen? For how long? In what medium? What was the time lapse between coculture initiation and ECP appearance?
Lines 107-109: This sentence describes the first step of the process and is nevertheless placed at the end of the paragraph. Please described the work in a logical order.
Tables 1 to 3: Why do the authors consider SFV from only two chimpanzee subspecies, while three have been described?
Description of viral promoters: The IP is described in the middle of the paragraph describing Env, in accordance to its location, but not to the biological function. Please describe IP in a paragraph following the description of LTR. Undescribed elements are: Tas responsive elements, splice donor site, the lack of central ppt, dimerization sequence, Poly A.
Lines 301-302: the location of the ER retention signal is erroneous.
Lines 295-302: Missing elements in the Env description are: The hydrophobic region in LP, the RBD in SU, and fusion peptide in TM.
Overall, the authors should refer to two papers (Schulze et al. 2011, and Rua et al., 2012) that had performed similar analysis that will help them to provide a comprehensive description of the new sequence.
Lines 339-346: Analyses of recombination. The authors should describe the sequences used for recombination analysis and the rational for their choice. We suggest that they expand the number of sequences. Why do the author limit their analysis to the concatamer? This strategy is expected to decrease the probability of detecting a recombination event.
SFV recombination has been described in two contexts. Some recombinant SFV infect a single or few NHPs/ Their parental strains are identified and prevalent in the studied NHP population (or their prey). The recombination is probably a recent event. The second context is an ancient recombination event that may have generated the bimorphic env genes found in feline FV and in SFV infecting chimpanzee, macaques, gorilla and AGM. The parental strains are unknown. This genetic diversity is functionally relevant as this region is targeted by neutralizing antibodies (Zemba et al., 2000; Lambert et al., 2018). Furthermore, env swapping is an important event in the biology and evolution of the retroviruses. The env gene fragments described as parental/recombinant in Galvin et al. (2013) or conserved/variant by Richard et al. (2015) should be aligned with those of SFV species comprising two genotypes. These alignment and phylogenic analysis will allow the authors to describe whether the SFVhpi env fragments are more related to one or the other 2 env genotypes from apes SFV.
Lines 450-451: “Comparison of LTR in the host and those obtained by culture are needed.” This sentence is erroneous, as the comparison has been reported in reference 56.
Specific comments
Line 27: “fills an important knowledge gap in SFV biology” sounds like an overstatement as no functional studies are presented in the paper. Please rephrase.
Lines 60-61: “the generation of database” is useless.
Lines 84-85: “The researchers were unable to obtain viral sequence” sounds awkward. First, it does not describe the experiment, but the people who carried the test; secondly word use deflates the contribution of the researchers. Please rephrase.
Author Response
Comments and Suggestions for Authors
Shankar et al. obtained the full length genetic sequence from an SFV strain isolated from pileated gibbon. They compared this sequence to those of other SFV species, analyzed the presence of functional motifs in genes and proteins and studied the evolutionary history of this strain. This thorough presentation of the first SFVhpi sequence represents significant knowledge for further basic virology and SFV diagnosis. The paper should be enhanced by a better description of recombination analysis, a study of env gene variability, rigorous checking of functional motifs, inclusion of recently obtained SFV sequences, adequate citation of the bibliography, and use of a more fluent writing style.
Broad comments
Line 35: “the functional SFV genome is double stranded DNA rather than single stranded-DNA”. This is an overstatement, as controversial data have been published. The two cited reviews state that DNA genome is functional but do not mention data on the relative infectivity of DNA and RNA genomes.
The evidence presented in Yu SF, Sullivan MD, Linial ML. Evidence that the human foamy virus genome is DNA. J Virol. 1999;73(2):1565–1572. (cited in the review by Pinto-Santini et al.) supports this fact that the particle-associated “DNA is sufficient for new rounds of replication” (Yu et al) . We do not discuss relative infectivity of DNA and RNA genomes, rather, we just state that the functional genome is double stranded DNA. Nonetheless, we have removed the word “functional” from this sentence.
Lines 34 and 37: two redundant sentences are separated by a third one. Please edit this paragraph and the whole manuscript for fluency.
We have checked these sentences and cannot find the redundancies the reviewer cites. We have addressed areas we feel are appropriate and as suggested elsewhere by this and the other reviewers.
Lines 41-43: For the description of SFV infection in NHPs, the authors cite only their papers, but neither a review, nor major papers from other teams. Lines 43-51: For the description of SFV infection in humans, the authors cite their research papers and reviews, but no research papers from other teams. In addition, Ref 22 is erroneous as it described SFV infection in macaques and not in humans. To avoid skewing towards self-citation, the authors have to cite the most recent review(s) and research papers from all teams if published after the review.
The majority of work on SFV co-evolution has come from our team. Hence, those articles are naturally more citated. However, we now cite other SFV co-evolution articles, but not exhaustively, and include or substitute review articles as suggested.
Lines 50-53: Please cite more recent reviews for the description of in vitro cytopathic effect and of SFV infection in NHPs. The authors should add the two papers that search for medical consequences in humans, published by Boneva et al. in 2007 and Buseyne et al. in 2018.
We have added the relevant citations as suggested.
Lines 68-75: The authors provide the list of isolates with full length sequences provided in ref 28. Additional complete sequences are available in Genbank, including SFV from Papio Anubis, and from Brachyteles arachnoides. These should be used in the analysis.
Since we already have four representatives for SFV from New World monkeys, including a spider monkey species (Ateles geoffroyi), as well as representatives from other Old World monkeys, to provide a diverse data set for comparison, we feel that addition of these additional sequences to the analysis for the purpose of this manuscript, will not change the FV phylogeny or recombination results given the co-evolutionary history of FVs. For example, we also have a complete baboon SFV genome (GenBank accession number MF472626) that we included in the analyses and which did not change the results presented in our manuscript. We removed the baboon SFV from the paper since we plan to publish that sequence in detail elsewhere.
Lines 100-102: The treatment of PBMCs need to be described. Where the cells fresh? Frozen? Stimulated with a mitogen? For how long? In what medium? What was the time lapse between coculture initiation and ECP appearance?
Although this information is available in the cited reference, we briefly describe the culture details.
Lines 107-109: This sentence describes the first step of the process and is nevertheless placed at the end of the paragraph. Please described the work in a logical order.
We have changed the order as suggested.
Tables 1 to 3: Why do the authors consider SFV from only two chimpanzee subspecies, while three have been described?
We have used representative sequences from each group of primates for the purpose of comparative phylogeny rather than using all available sequences to minimize the computationally intensive phylogenetic analyses. We feel this approach provides sufficient molecular information for our study.
Description of viral promoters: The IP is described in the middle of the paragraph describing Env, in accordance to its location, but not to the biological function. Please describe IP in a paragraph following the description of LTR. Undescribed elements are: Tas responsive elements, splice donor site, the lack of central ppt, dimerization sequence, Poly A.
We have moved the paragraph as suggested. We also now add on page 8 the locations of the additional elements noted when they were identifiable.
Lines 301-302: the location of the ER retention signal is erroneous.
We provided the location of the ER RS using the TM codon positions and not the entire Env coding region. We have now added the Env codon positions to avoid confusion.
Lines 295-302: Missing elements in the Env description are: The hydrophobic region in LP, the RBD in SU, and fusion peptide in TM.
Overall, the authors should refer to two papers (Schulze et al. 2011, and Rua et al., 2012) that had performed similar analysis that will help them to provide a comprehensive description of the new sequence.
Thanks for the suggestion. We used both papers for identifying motifs in our original submission. However, neither the Schulze nor the Rua articles show these additional motifs. Rua describes the hydrophobic tryptophan (W) residues being in the first 15 amino acids, which differs from that of Lindeman and Goepfert CTMI 2003 chapter) which shows them at pos 86/86 of LP (SP). The WXXW motif in SFVhpi is also within the first 15 amino acids at positions 10-13. In comparison to PFV we located a putative RBD in SU and the fusion peptide in TM. We have added this information to the results.
Lines 339-346: Analyses of recombination. The authors should describe the sequences used for recombination analysis and the rational for their choice. We suggest that they expand the number of sequences. Why do the author limit their analysis to the concatamer? This strategy is expected to decrease the probability of detecting a recombination event.
SFV recombination has been described in two contexts. Some recombinant SFV infect a single or few NHPs/ Their parental strains are identified and prevalent in the studied NHP population (or their prey). The recombination is probably a recent event. The second context is an ancient recombination event that may have generated the bimorphic env genes found in feline FV and in SFV infecting chimpanzee, macaques, gorilla and AGM. The parental strains are unknown. This genetic diversity is functionally relevant as this region is targeted by neutralizing antibodies (Zemba et al., 2000; Lambert et al., 2018). Furthermore, env swapping is an important event in the biology and evolution of the retroviruses. The env gene fragments described as parental/recombinant in Galvin et al. (2013) or conserved/variant by Richard et al. (2015) should be aligned with those of SFV species comprising two genotypes. These alignment and phylogenic analysis will allow the authors to describe whether the SFVhpi env fragments are more related to one or the other 2 env genotypes from apes SFV.
Please see lines 154-162 of Methods where we describe the sequences used and the rationale for using a concatamer. Briefly, we used the concatamer in order to maximize the molecular information across the coding regions with the most similarity to other SFV, while also ensuring a high quality alignment. This approach is common in the analysis of slow evolving cell-associated retroviruses. Using the entire genome creates alignment challenges because of highly divergent regions of the genome, which don’t align well and are then gap-stripped using SimPlot to evaluate recombination. Furthermore, recombination occurs in areas of homology and not high divergence and has mostly been reported in env, gag, and RT of closely related species. We have completed the additional recombination analyses using the sequences from the Galvin and Richard articles as suggested and have added this information to the paper. These analyses still did not show recombination in env for SFVhpi.
Lines 450-451: “Comparison of LTR in the host and those obtained by culture are needed.” This sentence is erroneous, as the comparison has been reported in reference 56.
Sorry for any confusion. Our statement refers to the current study where these comparisons of the SFVhpi LTRs are still needed but which are beyond the scope of the current study.
Specific comments
Line 27: “fills an important knowledge gap in SFV biology” sounds like an overstatement as no functional studies are presented in the paper. Please rephrase.
We used biology in a general sense, since the term encompasses a wide array of information. We have changed it to genetics.
Lines 60-61: “the generation of database” is useless.
We have re-phrased the sentence to read “The design of molecular methods for the accurate identification of SFV infection requires a database containing representative sequences from divergent SFV lineages”.
Lines 84-85: “The researchers were unable to obtain viral sequence” sounds awkward. First, it does not describe the experiment, but the people who carried the test; secondly word use deflates the contribution of the researchers. Please rephrase.
We have re-phrased the sentence as “A recent study reported a high seroprevalence of SFV in gibbons from Cambodia although the seropositive samples were not SFV PCR positive”.
Reviewer 2 Report
Dear authors,
it was a pleasure to read this manuscript, the work is solid and the data are well presented.
My only suggestion would be to alter the sentence about "disease" in the abstract. I dont think the full Genome is mainly of interest to detect potential disease causing properties, this full genome study is interesting as such.
Author Response
Comments and Suggestions for Authors
Dear authors,
it was a pleasure to read this manuscript, the work is solid and the data are well presented.
My only suggestion would be to alter the sentence about "disease" in the abstract. I dont think the full Genome is mainly of interest to detect potential disease causing properties, this full genome study is interesting as such.
This sentence has been modified to read, “SFVs can zoonotically infect humans but very few complete SFV genomes are available, hampering the design of diagnostic assays”.
Reviewer 3 Report
The manuscript by Shankar et al describes the first complete genome analysis simian foamy virus from pileated gibbon ape (Hylobates pileatus) inferring the co-evolution of host and virus. The work presented is straightforward. It is based on sequencing and sequence analysis to determine host virus evolution as well as sequence comparisons with other foamy virus strains and description of putative motifs based on sequence. Although this paper describes the first complete sequence foamy virus from Hylobates pileatusthe main finding of co-evolution of foamy virus with the host is not new or novel except strengthen the previous studies. Additional new foamy virus genome sequence, however, provide information for further understanding the epidemiology and evolutionary history of foamy viruses.
Other points
1. Sequence analysis show that the genome is the longest of foamy viruses sequenced thus far due the a much longer LTR. However, caution should be taken because older sequence analyses show insertion and deletion in the LTR within the prototypye foamy virus even at times showing differences 5’ LTR from the 3’ LTR.
2. The genome is sequenced after growing the virus in cf2th cell line, which would be under no host influence. Would this lead to sequence variation that would impede on the conclusion of co-evolving with the host?
3. The sentence page 1, lines 37-38 “To initiate infection, FVs use multiple promoters, a trait seen in complex DNA viruses, but not in other retroviruses [3].” is awkward and needs re-writing.
Author Response
Comments and Suggestions for Authors
The manuscript by Shankar et al describes the first complete genome analysis simian foamy virus from pileated gibbon ape (Hylobates pileatus) inferring the co-evolution of host and virus. The work presented is straightforward. It is based on sequencing and sequence analysis to determine host virus evolution as well as sequence comparisons with other foamy virus strains and description of putative motifs based on sequence. Although this paper describes the first complete sequence foamy virus from Hylobates pileatus the main finding of co-evolution of foamy virus with the host is not new or novel except strengthen the previous studies. Additional new foamy virus genome sequence, however, provide information for further understanding the epidemiology and evolutionary history of foamy viruses.
Other points
Sequence analysis show that the genome is the longest of foamy viruses sequenced thus far due the a much longer LTR. However, caution should be taken because older sequence analyses show insertion and deletion in the LTR within the prototypye foamy virus even at times showing differences 5’ LTR from the 3’ LTR.
We have confirmed the length of the LTR using SFVhpi-specific primers and PCR with traditional Sanger sequencing and have added this information to the manuscript.
2. The genome is sequenced after growing the virus in cf2th cell line, which would be under no host influence. Would this lead to sequence variation that would impede on the conclusion of co-evolving with the host?
Foamy viruses are extremely stable in vivo and in vitro. Our phylogenetic and recombination analyses show that culturing in Cf2th cells does not have an effect on the way SFVhpi sequence clusters in relation to FVs from other species. The position of SFVhpi in the tree constructed from viral sequence alignments almost exactly mirrors that of the phylogeny from host mitochondrial sequences, indicating co-evolution. In addition, the pol sequence from the PBMC and Cf2Th DNAs, although short (425-bp) are 100% identical supporting the stability. To adequately test this hypothesis would require obtaining the complete genome from the PBMC DNA, which is beyond the scope of our study.
3. The sentence page 1, lines 37-38 “To initiate infection, FVs use multiple promoters, a trait seen in complex DNA viruses, but not in other retroviruses [3].” is awkward and needs re-writing.
This sentence has been re-worded to read, “FVs also differ from other retroviruses in how they initiate infection. Like complex DNA viruses, they use multiple promoters [4]."
Round 2
Reviewer 1 Report
See attached file
Comments and Suggestions for Authors
Shankar et al. obtained the full length genetic sequence from an SFV strain isolated from pileated gibbon. They compared this sequence to those of other SFV species, analyzed the presence of functional motifs in genes and proteins and studied the evolutionary history of this strain. This thorough presentation of the first SFVhpi sequence represents significant knowledge for further basic virology and SFV diagnosis. The paper should be enhanced by a better description of recombination analysis, a study of env gene variability, rigorous checking of functional motifs, inclusion of recently obtained SFV sequences, adequate citation of the bibliography, and use of a more fluent writing style.
Second review: Shankar et al. improved their paper. Some elements of their rebuttal are still not convincing.
Broad comments
Line 35: “the functional SFV genome is double stranded DNA rather than single stranded-DNA”. This is an overstatement, as controversial data have been published. The two cited reviews state that DNA genome is functional but do not mention data on the relative infectivity of DNA and RNA genomes.
The evidence presented in Yu SF, Sullivan MD, Linial ML. Evidence that the human foamy virus genome is DNA. J Virol. 1999;73(2):1565–1572. (cited in the review by Pinto-Santini et al.) supports this fact that the particle-associated “DNA is sufficient for new rounds of replication” (Yu et al) . We do not discuss relative infectivity of DNA and RNA genomes, rather, we just state that the functional genome is double stranded DNA. Nonetheless, we have removed the word “functional” from this sentence.
SFV genome can be DNA or RNA. The data from Yu et al. have not been confirmed by others.
Lines 34 and 37: two redundant sentences are separated by a third one. Please edit this paragraph and the whole manuscript for fluency.
We have checked these sentences and cannot find the redundancies the reviewer cites. We have addressed areas we feel are appropriate and as suggested elsewhere by this and the other reviewers.
Here they are:
For example, FVs use two promoters for gene expression,one in the long terminal repeat (LTR) and one in the envelope (env) gene. In addition, FVs complete reverse transcription within the virion before infection of a new host cell, such that the functional SFV genome is double-stranded DNA rather than single-stranded RNA [1, 2]. To initiate infection, FVs use multiple promoters, a trait seen in complex DNA viruses, but not in other retroviruses [3].
Lines 41-43: For the description of SFV infection in NHPs, the authors cite only their papers, but neither a review, nor major papers from other teams. Lines 43-51: For the description of SFV infection in humans, the authors cite their research papers and reviews, but no research papers from other teams. In addition, Ref 22 is erroneous as it described SFV infection in macaques and not in humans. To avoid skewing towards self-citation, the authors have to cite the most recent review(s) and research papers from all teams if published after the review.
The majority of work on SFV co-evolution has come from our team. Hence, those articles are naturally more citated. However, we now cite other SFV co-evolution articles, but not exhaustively, and include or substitute review articles as suggested.
There are no change in the bibliography. The authors have not address the whole comment.
Lines 50-53: Please cite more recent reviews for the description of in vitro cytopathic effect and of SFV infection in NHPs. The authors should add the two papers that search for medical consequences in humans, published by Boneva et al. in 2007 and Buseyne et al. in 2018.
We have added the relevant citations as suggested.
One citation is missing
Lines 68-75: The authors provide the list of isolates with full length sequences provided in ref 28. Additional complete sequences are available in Genbank, including SFV from Papio Anubis, and from Brachyteles arachnoides. These should be used in the analysis.
Since we already have four representatives for SFV from New World monkeys, including a spider monkey species (Ateles geoffroyi), as well as representatives from other Old World monkeys, to provide a diverse data set for comparison, we feel that addition of these additional sequences to the analysis for the purpose of this manuscript, will not change the FV phylogeny or recombination results given the co-evolutionary history of FVs. For example, we also have a complete baboon SFV genome (GenBank accession number MF472626) that we included in the analyses and which did not change the results presented in our manuscript. We removed the baboon SFV from the paper since we plan to publish that sequence in detail elsewhere.
My comment was on the citation of available full length SFV sequence, not on the phylogeny or recombination analysis. The text can be easily updated.
Lines 100-102: The treatment of PBMCs need to be described. Where the cells fresh? Frozen? Stimulated with a mitogen? For how long? In what medium? What was the time lapse between coculture initiation and ECP appearance?
Although this information is available in the cited reference, we briefly describe the culture details.
OK
Lines 107-109: This sentence describes the first step of the process and is nevertheless placed at the end of the paragraph. Please described the work in a logical order.
We have changed the order as suggested.
OK
Tables 1 to 3: Why do the authors consider SFV from only two chimpanzee subspecies, while three have been described?
We have used representative sequences from each group of primates for the purpose of comparative phylogeny rather than using all available sequences to minimize the computationally intensive phylogenetic analyses. We feel this approach provides sufficient molecular information for our study.
Description of viral promoters: The IP is described in the middle of the paragraph describing Env, in accordance to its location, but not to the biological function. Please describe IP in a paragraph following the description of LTR. Undescribed elements are: Tas responsive elements, splice donor site, the lack of central ppt, dimerization sequence, Poly A.
We have moved the paragraph as suggested. We also now add on page 8 the locations of the additional elements noted when they were identifiable.
OK
Lines 301-302: the location of the ER retention signal is erroneous.
We provided the location of the ER RS using the TM codon positions and not the entire Env coding region. We have now added the Env codon positions to avoid confusion.
OK
Lines 295-302: Missing elements in the Env description are: The hydrophobic region in LP, the RBD in SU, and fusion peptide in TM.
Overall, the authors should refer to two papers (Schulze et al. 2011, and Rua et al., 2012) that had performed similar analysis that will help them to provide a comprehensive description of the new sequence.
Thanks for the suggestion. We used both papers for identifying motifs in our original submission. However, neither the Schulze nor the Rua articles show these additional motifs. Rua describes the hydrophobic tryptophan (W) residues being in the first 15 amino acids, which differs from that of Lindeman and Goepfert CTMI 2003 chapter) which shows them at pos 86/86 of LP (SP). The WXXW motif in SFVhpi is also within the first 15 amino acids at positions 10-13. In comparison to PFV we located a putative RBD in SU and the fusion peptide in TM. We have added this information to the results.
I do agree that the two papers (Schulze et al. 2011, and Rua et al., 2012) do not describe all motifs. I proposed them as model to describe new sequences.
Lines 339-346: Analyses of recombination. The authors should describe the sequences used for recombination analysis and the rational for their choice. We suggest that they expand the number of sequences. Why do the author limit their analysis to the concatamer? This strategy is expected to decrease the probability of detecting a recombination event.
SFV recombination has been described in two contexts. Some recombinant SFV infect a single or few NHPs/ Their parental strains are identified and prevalent in the studied NHP population (or their prey). The recombination is probably a recent event. The second context is an ancient recombination event that may have generated the bimorphic env genes found in feline FV and in SFV infecting chimpanzee, macaques, gorilla and AGM. The parental strains are unknown. This genetic diversity is functionally relevant as this region is targeted by neutralizing antibodies (Zemba et al., 2000; Lambert et al., 2018). Furthermore, env swapping is an important event in the biology and evolution of the retroviruses. The env gene fragments described as parental/recombinant in Galvin et al. (2013) or conserved/variant by Richard et al. (2015) should be aligned with those of SFV species comprising two genotypes. These alignment and phylogenic analysis will allow the authors to describe whether the SFVhpi env fragments are more related to one or the other 2 env genotypes from apes SFV.
Please see lines 154-162 of Methods where we describe the sequences used and the rationale for using a concatamer. Briefly, we used the concatamer in order to maximize the molecular information across the coding regions with the most similarity to other SFV, while also ensuring a high quality alignment. This approach is common in the analysis of slow evolving cell-associated retroviruses. Using the entire genome creates alignment challenges because of highly divergent regions of the genome, which don’t align well and are then gap-stripped using SimPlot to evaluate recombination. Furthermore, recombination occurs in areas of homology and not high divergence and has mostly been reported in env, gag, and RT of closely related species. We have completed the additional recombination analyses using the sequences from the Galvin and Richard articles as suggested and have added this information to the paper. These analyses still did not show recombination in env for SFVhpi.
Ok
Lines 450-451: “Comparison of LTR in the host and those obtained by culture are needed.” This sentence is erroneous, as the comparison has been reported in reference 56.
Sorry for any confusion. Our statement refers to the current study where these comparisons of the SFVhpi LTRs are still needed but which are beyond the scope of the current study.
OK
Specific comments
Line 27: “fills an important knowledge gap in SFV biology” sounds like an overstatement as no functional studies are presented in the paper. Please rephrase.
We used biology in a general sense, since the term encompasses a wide array of information. We have changed it to genetics.
OK
Lines 60-61: “the generation of database” is useless.
We have re-phrased the sentence to read “The design of molecular methods for the accurate identification of SFV infection requires a database containing representative sequences from divergent SFV lineages”.
OK
Lines 84-85: “The researchers were unable to obtain viral sequence” sounds awkward. First, it does not describe the experiment, but the people who carried the test; secondly word use deflates the contribution of the researchers. Please rephrase.
We have re-phrased the sentence as “A recent study reported a high seroprevalence of SFV in gibbons from Cambodia although the seropositive samples were not SFV PCR positive”.

Author Response
Reviewer 1:
Comments and Suggestions for Authors
Shankar et al. obtained the full length genetic sequence from an SFV strain isolated from
pileated gibbon. They compared this sequence to those of other SFV species, analyzed the
presence of functional motifs in genes and proteins and studied the evolutionary history of
this strain. This thorough presentation of the first SFVhpi sequence represents significant
knowledge for further basic virology and SFV diagnosis. The paper should be enhanced by a
better description of recombination analysis, a study of env gene variability, rigorous
checking of functional motifs, inclusion of recently obtained SFV sequences, adequate
citation of the bibliography, and use of a more fluent writing style.
Second review: Shankar et al. improved their paper. Some elements of their rebuttal are still not
convincing.
Thanks for agreeing that the manuscript was improved. We hope the new revisions are satisfactory.
Broad comments
Line 35: “the functional SFV genome is double stranded DNA rather than single stranded-
DNA”. This is an overstatement, as controversial data have been published. The two cited
reviews state that DNA genome is functional but do not mention data on the relative
infectivity of DNA and RNA genomes.
The evidence presented in Yu SF, Sullivan MD, Linial ML. Evidence that the human foamy
virus genome is DNA. J Virol. 1999;73(2):1565–1572. (cited in the review by Pinto-Santini
et al.) supports this fact that the particle-associated “DNA is sufficient for new rounds of
replication” (Yu et al) . We do not discuss relative infectivity of DNA and RNA genomes,
rather, we just state that the functional genome is double stranded DNA. Nonetheless, we
have removed the word “functional” from this sentence.
SFV genome can be DNA or RNA. The data from Yu et al. have not been confirmed by
others.
We have changed this sentence to state the genome can be DNA or RNA.
Lines 34 and 37: two redundant sentences are separated by a third one. Please edit this
paragraph and the whole manuscript for fluency.
We have checked these sentences and cannot find the redundancies the reviewer cites. We
have addressed areas we feel are appropriate and as suggested elsewhere by this and the other
reviewers.
Here they are:
For example, FVs use two promoters for gene expression,one in the long terminal repeat
(LTR) and one in the envelope (env) gene. In addition, FVs complete reverse transcription
within the virion before infection of a new host cell, such that the functional SFV genome is
double-stranded DNA rather than single-stranded RNA [1, 2]. To initiate infection, FVs use
multiple promoters, a trait seen in complex DNA viruses, but not in other retroviruses [3].
Thanks. We have removed the redundant multiple promoters information.
Lines 41-43: For the description of SFV infection in NHPs, the authors cite only their papers,
but neither a review, nor major papers from other teams. Lines 43-51: For the description of
SFV infection in humans, the authors cite their research papers and reviews, but no research
papers from other teams. In addition, Ref 22 is erroneous as it described SFV infection in
macaques and not in humans. To avoid skewing towards self-citation, the authors have to cite
the most recent review(s) and research papers from all teams if published after the review.
The majority of work on SFV co-evolution has come from our team. Hence, those articles are
naturally more citated. However, we now cite other SFV co-evolution articles, but not
exhaustively, and include or substitute review articles as suggested.
There are no change in the bibliography. The authors have not address the whole comment.
Thanks. This was an Endnote formatting issue. The original manuscript had 63 references whereas the revision has 76.
Lines 50-53: Please cite more recent reviews for the description of in vitro cytopathic effect
and of SFV infection in NHPs. The authors should add the two papers that search for medical
consequences in humans, published by Boneva et al. in 2007 and Buseyne et al. in 2018.
We have added the relevant citations as suggested.
One citation is missing
This may also have been an Endnote formatting issue. Both citations were included in the first revision as references 31, and 32.
Lines 68-75: The authors provide the list of isolates with full length sequences provided in ref
28. Additional complete sequences are available in Genbank, including SFV from Papio
Anubis, and from Brachyteles arachnoides. These should be used in the analysis.
Since we already have four representatives for SFV from New World monkeys, including a
spider monkey species (Ateles geoffroyi), as well as representatives from other Old World
monkeys, to provide a diverse data set for comparison, we feel that addition of these
additional sequences to the analysis for the purpose of this manuscript, will not change the FV
phylogeny or recombination results given the co-evolutionary history of FVs. For example,
we also have a complete baboon SFV genome (GenBank accession number MF472626) that
we included in the analyses and which did not change the results presented in our manuscript.
We removed the baboon SFV from the paper since we plan to publish that sequence in detail
elsewhere.
My comment was on the citation of available full length SFV sequence, not on the phylogeny
or recombination analysis. The text can be easily updated.
We are confused by the reviewer’s new request. Clearly, they requested these additional SFVs be included in the “analysis”. There is no need to cite these additional SFVs if they were not included in the analysis.
Lines 100-102: The treatment of PBMCs need to be described. Where the cells fresh? Frozen?
Stimulated with a mitogen? For how long? In what medium? What was the time lapse
between coculture initiation and ECP appearance?
Although this information is available in the cited reference, we briefly describe the culture
details.
OK
Lines 107-109: This sentence describes the first step of the process and is nevertheless placed
at the end of the paragraph. Please described the work in a logical order.
We have changed the order as suggested.
OK
Tables 1 to 3: Why do the authors consider SFV from only two chimpanzee subspecies, while
three have been described?
We have used representative sequences from each group of primates for the purpose of
comparative phylogeny rather than using all available sequences to minimize the
computationally intensive phylogenetic analyses. We feel this approach provides sufficient
molecular information for our study.
Description of viral promoters: The IP is described in the middle of the paragraph describing
Env, in accordance to its location, but not to the biological function. Please describe IP in a
paragraph following the description of LTR. Undescribed elements are: Tas responsive
elements, splice donor site, the lack of central ppt, dimerization sequence, Poly A.
We have moved the paragraph as suggested. We also now add on page 8 the locations of the
additional elements noted when they were identifiable.
OK
Lines 301-302: the location of the ER retention signal is erroneous.
We provided the location of the ER RS using the TM codon positions and not the entire Env
coding region. We have now added the Env codon positions to avoid confusion.
OK
Lines 295-302: Missing elements in the Env description are: The hydrophobic region in LP,
the RBD in SU, and fusion peptide in TM.
Overall, the authors should refer to two papers (Schulze et al. 2011, and Rua et al., 2012) that
had performed similar analysis that will help them to provide a comprehensive description of
the new sequence.
Thanks for the suggestion. We used both papers for identifying motifs in our original
submission. However, neither the Schulze nor the Rua articles show these additional motifs.
Rua describes the hydrophobic tryptophan (W) residues being in the first 15 amino acids,
which differs from that of Lindeman and Goepfert CTMI 2003 chapter) which shows them at
pos 86/86 of LP (SP). The WXXW motif in SFVhpi is also within the first 15 amino acids at
positions 10-13. In comparison to PFV we located a putative RBD in SU and the fusion
peptide in TM. We have added this information to the results.
I do agree that the two papers (Schulze et al. 2011, and Rua et al., 2012) do not describe all
motifs. I proposed them as model to describe new sequences.
Thanks. We have already made the changes in the first revision.
Lines 339-346: Analyses of recombination. The authors should describe the sequences used
for recombination analysis and the rational for their choice. We suggest that they expand the
number of sequences. Why do the author limit their analysis to the concatamer? This strategy
is expected to decrease the probability of detecting a recombination event.
SFV recombination has been described in two contexts. Some recombinant SFV infect a
single or few NHPs/ Their parental strains are identified and prevalent in the studied NHP
population (or their prey). The recombination is probably a recent event. The second context
is an ancient recombination event that may have generated the bimorphic env genes found in
feline FV and in SFV infecting chimpanzee, macaques, gorilla and AGM. The parental strains
are unknown. This genetic diversity is functionally relevant as this region is targeted by
neutralizing antibodies (Zemba et al., 2000; Lambert et al., 2018). Furthermore, env swapping
is an important event in the biology and evolution of the retroviruses. The env gene fragments
described as parental/recombinant in Galvin et al. (2013) or conserved/variant by Richard et
al. (2015) should be aligned with those of SFV species comprising two genotypes. These
alignment and phylogenic analysis will allow the authors to describe whether the SFVhpi env
fragments are more related to one or the other 2 env genotypes from apes SFV.
Please see lines 154-162 of Methods where we describe the sequences used and the rationale
for using a concatamer. Briefly, we used the concatamer in order to maximize the molecular
information across the coding regions with the most similarity to other SFV, while also
ensuring a high quality alignment. This approach is common in the analysis of slow evolving
cell-associated retroviruses. Using the entire genome creates alignment challenges because of
highly divergent regions of the genome, which don’t align well and are then gap-stripped
using SimPlot to evaluate recombination. Furthermore, recombination occurs in areas of
homology and not high divergence and has mostly been reported in env, gag, and RT of
closely related species. We have completed the additional recombination analyses using the
sequences from the Galvin and Richard articles as suggested and have added this information
to the paper. These analyses still did not show recombination in env for SFVhpi.
Ok
Lines 450-451: “Comparison of LTR in the host and those obtained by culture are needed.”
This sentence is erroneous, as the comparison has been reported in reference 56.
Sorry for any confusion. Our statement refers to the current study where these comparisons of
the SFVhpi LTRs are still needed but which are beyond the scope of the current study.
OK
Specific comments
Line 27: “fills an important knowledge gap in SFV biology” sounds like an overstatement as
no functional studies are presented in the paper. Please rephrase.
We used biology in a general sense, since the term encompasses a wide array of information.
We have changed it to genetics.
OK
Lines 60-61: “the generation of database” is useless.
We have re-phrased the sentence to read “The design of molecular methods for the accurate
identification of SFV infection requires a database containing representative sequences from
divergent SFV lineages”.
OK
Lines 84-85: “The researchers were unable to obtain viral sequence” sounds awkward. First, it
does not describe the experiment, but the people who carried the test; secondly word use
deflates the contribution of the researchers. Please rephrase.
We have re-phrased the sentence as “A recent study reported a high seroprevalence of SFV in
gibbons from Cambodia although the seropositive samples were not SFV PCR positive”.
The reviewer did not provide comments on our revision.